# Evaluation of Marker Materials and Spectroscopic Methods for Tracer-Based Sorting of Plastic Wastes

**DOI:** 10.3390/polym14153074

**Published:** 2022-07-29

**Authors:** Christoph Olscher, Aleksander Jandric, Christian Zafiu, Florian Part

**Affiliations:** 1Department of Water-Atmosphere-Environment, Institute of Waste Management, University of Natural Resources and Life Sciences, Muthgasse 107, 1190 Vienna, Austria; christoph.olscher@boku.ac.at (C.O.); christian.zafiu@boku.ac.at (C.Z.); florian.part@boku.ac.at (F.P.); 2Bundesanstalt für Materialforschung und -Prüfung (BAM), 3.1 Fachbereich Gefahrgutverpackungen, Unter den Eichen, 44-46, 12203 Berlin, Germany

**Keywords:** Circular Economy, post-consumer plastic waste, plastic waste recycling, tracer-based sorting, sensor-based sorting, spectroscopy, fluorescent markers, thermoplastics, polyoxymethylene

## Abstract

Plastics are a ubiquitous material with good mechanical, chemical and thermal properties, and are used in all industrial sectors. Large quantities, widespread use, and insufficient management of plastic wastes lead to low recycling rates. The key challenge in recycling plastic waste is achieving a higher degree of homogeneity between the different polymer material streams. Modern waste sorting plants use automated sensor-based sorting systems capable to sort out commodity plastics, while many engineering plastics, such as polyoxymethylene (POM), will end up in mixed waste streams and are therefore not recycled. A novel approach to increasing recycling rates is tracer-based sorting (TBS), which uses a traceable plastic additive or marker that enables or enhances polymer type identification based on the tracer’s unique fingerprint (e.g., fluorescence). With future TBS applications in mind, we have summarized the literature and assessed TBS techniques and spectroscopic detection methods. Furthermore, a comprehensive list of potential tracer substances suitable for thermoplastics was derived from the literature. We also derived a set of criteria to select the most promising tracer candidates (3 out of 80) based on their material properties, toxicity profiles, and detectability that could be applied to enable the circularity of, for example, POM or other thermoplastics.

## 1. Introduction

Plastic materials have become ubiquitous in modern societies, due to their versatile mechanical properties and low production costs. Global plastic production increased from 230 to 367 (approx. +60%) million tons since 2005 [1,2]. In the European Union (EU27 including Norway, Switzerland, and United Kingdom), the largest end-use market is the packaging sector (40.5%), followed by building and construction (20.4%), automotive (8.8%), electrical and electronic equipment (6.2%), and others (24.1%) [2]. However, Geyer et al. estimated that 60% of the whole global production of plastics between 1950–2015 were discarded and accumulated in landfills or the environment [3]. This plastic is mostly lost for recycling and causes macro- and microplastic pollution found today in agricultural soils [4], in the Arctic sea ice, or even in human placenta [5], highlighting the importance of efficient plastic waste collection and recycling.

Besides the challenge to adequately and efficiently collect plastic waste, mechanical or chemical recycling is imperative to decrease the greenhouse gas impacts along the lifecycle of plastic products and wastes [6]. In 2020 in the EU (EU27 +3), 10.2 million tons (ca. 34.5%) of the post-consumer plastic waste were recycled out of 29.5 million tons collected [2]. Of these, by far the largest share of plastic recycled originates from the packaging sector, where the average recycling rate is approximately 41% [7], while the recycling rate of plastic waste from all other sectors (building and construction, automotive, electric and electronic, etc.) is significantly lower [8]. For this reason, in 2020, the Circular Economy Action Plan of the EU Commission [9] was revised, and treatment and monitoring of plastic waste was assigned a central role, together with restrictions for adding (primary) microplastics to products, labeling of plastic materials, and increasing the overall share of recycled plastics. Furthermore, in the EU Directive 2012/19/EU on Waste Electrical and Electronic Equipment (WEEE) recycling targets between 55–80 wt.% over all materials were specified, that are impossible to meet without increasing the recycling of the plastic components from Electrical and Electronic Equipment (EEE) [10]. The biggest technical challenge is that collected plastic waste is in general a heterogenic mix of different polymer types as well as organic, inorganic, and metal contaminants, depending on the disposal and collecting scheme [11,12,13]. The largest polymer streams are polyethylene (PE) types—i.e., high-density polyethylene (PE-HD), low-density polyethylene (PE-LD), linear-low-density polyethylene (PE-LLD), middle-density polyethylene (PE-MD), polypropylene (PP), polystyrene (PS), polystyrene-expendable (PS-E), and polyvinylchloride (PVC), which are mainly used for the packaging, construction, automotive or electric and electronic sector [2].

The fundamental prerequisite for recycling plastics is the homogeneity of the input material, as different polymers have distinctive chemical and mechanical properties and therefore cannot be mixed with each other arbitrarily, as shown in the Appendix A [14,15]. Within the recycling process, almost any divergence is considered as an impurity and leads to a loss of quality of the recycled material [11]. Separate collection by polymer type or plastic product (e.g., for PET bottles) or mechanical pre-treatment are measures among many to consequently improve quality of input materials or feedstock for mechanical, chemical, enzymatic, or thermal recycling of plastics [13]. The current state-of-the-art automatic sorting systems for pre-treatment and plastic separation are capable of sorting different bulk plastics and may very quickly reach their operational limit if the input material stream is too heterogeneous in terms of size, shape, and material quality. For example, in current state-of-the-art recycling plants for separately collected lightweight packaging materials, valuable plastic fractions of polyethylene terephthalate (PET) bottles, polyolefin (PO) agglomerates, polystyrene (PS) granulates, or polyethylene (PE) granulates are separated using a combination of air separators, centrifuges, float-sink processes, and near infrared (NIR) sorters [16,17]. The material composition of plastics from separately collected waste of electrical and electronic equipment is more complex, as shown, for example, by Jandric et al., in the case of small household appliances where 18 different polymer types could be identified by Fourier-transform infrared spectroscopy (FT-IR). During WEEE recycling, polymers are sorted out by float-sink processes and air separators, to homogenize the commodity plastics from this waste stream, such as acrylonitrile butadiene styrene (ABS), PP, and PS. Considering all plastic flows and the different polymer types, a sorting efficiency of >95% is only possible to a very limited extent, even for elaborately pre-treated input material [17,18,19,20]. In praxis, specific sorting processes that target distinct polymers are not applied to polymers that occur in small quantities and/or exhibit a complex material composition (e.g., polymer composites). Therefore, polymers, such as polyoxymethylene (POM), end up in the mixed plastic material stream during recycling, which are incinerated and thus not recycled.

Labeling of different plastic components to enable an easier identification of the polymeric composition was seen as the next logical step to improve sorting efficiency. As a result, plastic components have been labeled with the so-called resin identification or recycling code (RIC) which is imprinted and allows a visual identification of the polymer. The first RIC was established in the late 1980s by the American Society for Testing and Materials (ASTM) Regulation D7611 [21]. In the EU, the labeling of plastic polymer products is regulated in the DIN EN ISO 11469:2016 and the technical terms are specified in ISO 1043 Part 1–4 [22,23,24,25,26]. However, such imprinted labels for single components are not useful during mechanical plastic recycling, such as shredding, because the label might become lost or destroyed during the process and cannot be detected by automated sorting techniques using optical sensors or cameras. Therefore, RICs are only useful in cases of manual disassembly and sorting. An approach that improves automatic identification and separation is to incorporate distinct plastic additives that add unique detectable features to the plastics during production and allow to increase the sorting efficiency and consequently save CO_2_ emissions by yielding high quality recycled plastics [27]. This technique is also known as marker- or tracer-based sorting (TBS), which is based on marker materials that enable their detection due to unique fingerprints (e.g., fluorescence-based inorganic markers improving sensor-based sorting) [28,29]. The first applications of using markers in polymers were already made in the early 1990s [28,30,31,32,33]. A recent case study on lightweight packaging materials in Germany shows that TBS technology can save up to 1227 kg CO_2_-eq, compared to conventional sorting and recycling [27]. For this, the tracers are either dispersed and directly embedded into the polymer matrix, or tracer-based inks are printed onto a product label. One or more substances, such as lanthanoid-based complexes, are used as the markers for TBS that are detectable by up-conversion photoluminescence or other spectroscopic detection techniques [28,30]. It is also possible to incorporate multiple tracers in a polymer to provide multiple fingerprints for detection [29,31]. The establishment of such a TBS-based coding system enables a sorting technology that would meet the requirements to identify and separate the large variety of different polymers, polymer composites, or highly contaminated polymers in heterogenic waste streams [18,29].

Distinguished from the commodity plastics (PE, PO, PET, etc.), there is another subgroup of thermoplastic polymers called engineering plastics, consisting of POM, Poly(methyl methacrylate) (PMMA), polycarbonates (PC), polybutylene terephthalate (PBT), and other polymer types that contain special fillers and additives. They have modified mechanical properties, such as greater impact strength, higher elasticity, durability, etc. These properties allow for broader application (e.g., for the construction or avionic sector), but due to the comparatively higher costs to other thermoplastics, their quantity on the market is much lower. Among engineering plastics, POM is considered as a high-quality valuable polymer with a production volume of ca. 1.7 million tons per year with outstanding properties, such as high strength, hardness, and excellent dimensional stability and it is therefore widely used in the automobile industry, the construction sector, or as part of household items like grinders, clips, window fittings, gears for printers, etc. [34,35,36,37]. However, during recycling and extrusion of thermosets, POM is considered as interfering material because of its specific molecular structure and crystallinity, that makes it thermodynamically immiscible and can therefore influence material properties of the final product [14,38]. In addition, extrusion or regranulation processes of POM can lead to the formation of toxic formaldehyde (H_2_CO) gas and therefore a high share POM in other plastic waste stream (e.g., ABS from WEEE recycling) needs to be avoided. Despite the high market value and good recycling characteristics, as long as they are not mixed with other polymer types, plastics made of POM polymers are currently only marginally recycled. Therefore, the goal of this research is to develop a method for selecting the most promising tracer candidates suitable for future implementation of the TBS concept. In the case of POM, this approach would allow to create homogeneous POM streams and consequently, a closed loop recycling in high material quality. However, mixing tracer substances into polymer matrix during the production comes with risks, such as increased prices for the tracer, availability of the tracer material on the market, negative impact on the plastic properties, safety aspects, and additional regulatory processes. Most of these risks increase with the amount of tracer that must be introduced into the plastics, while a reasonable amount is determined by the limit of detection of the available detection technologies.

All of the mentioned parameters must be considered during the tracer selection process to find a practicable material may be accepted by producers, distributors, public authorities, and waste management companies. For this review article, the literature on TBS implementation studies was summarized to narrow down the list of potential tracers or markers for plastics, at which the corresponding analytical methods for their targeted detection as well as safety aspects were considered. Finally, the three most promising plastic marker candidates were selected based on market availability, processability in thermoplastics, potential toxicity, and detectability during recycling in line with the ‘safe and sustainable by design approach’ proposed by the European Commission [39].

## 2. Methodological Approach for Tracer Selection

### 2.1. Literature Research and Data Collection

In this study, the terms tracer and marker are used synonymously and are defined as a material, which includes chemicals, micro,- and nanoparticles, as well as colloids that have detectable properties such as isotopic, spectroscopic, or elemental fingerprints (e.g., ^13^C labels, rare earth metal dopants, semiconducting quantum dots, fluorescent dyes or pigments) that can be dispersed in the polymer matrix and facilitate detection and improve the efficiency of automated sensor-based sorting techniques. The selection of potential tracer candidates was based on a literature review on substances, particles, and pigments which can be easily detected by spectroscopic methods that can be, or already are implemented into sorting facilities for recyclables, municipal solid wastes, WEEE, and other polymer-containing waste streams. The literature was acquired through an online search using single or combined keywords (Appendix A) in the search engines ScienceDirect, PubMed, and Google Scholar. Aside from the resulting literature found by the search engines, a snowballing procedure was also applied to find more relevant marker materials and information on their application [40]. 

The result of the list of potential markers, provided in the Excel File Appendix A, is based on the methodological approach shown in Figure 1. Key performance indicators with regards to future implementation of the TBS concept were identified, such as market availability (indicated by the chemical abstract service (CAS) registry number), thermal properties, or potential toxicological hazards regarding the authorization of chemicals according to the registration, evaluation, authorization, and restriction of chemicals (REACH) regulation and the controlling, labeling and packaging (CLP) regulation [41]. Since different quantities and qualities of information are found for each substance, a hierarchy of “priority of information” was established, as highlighted in the yellow box in Figure 1. Information retrieved from REACH and CLP was regarded as the highest hierarchical level, followed by data from scientific databases (e.g., pubchem.ncbi.nlm.nih.gov) and data provided via safety data sheets and product datasheets. This prioritization of information was introduced to handle conflicting data, in which case the highest available level of data was used for the evaluation of the individual marker.

### 2.2. Selection Criteria for Polymer Tracers

To narrow down the selection of potential tracer candidates, a decision tree was developed based on key performance indicators shown in Figure 2. The selection criterion with the highest priority was the detection capability of a tracer. Regarding detection technologies, it is important to note that detection techniques are at different stages of industrial implementation or development, which is indicated by the Technology Readiness Level (TRL). The TRL is a concept, developed by NASA in the 1970s which provides 9 levels for the classification of the development state of a certain method/system [42]. In recent years, the usability of the TRL has spread into other industrial sectors and government bodies, and therefore guidelines were published for a harmonized TRL assessment [43,44]. It also needs to be noted that the methods discussed in Section 3 can have a different TRL in application areas outside plastic sorting, especially if they have not been originally developed for recycling. The presented TRL was assessed based on the review of literature on spectroscopic methods to sort polymers and is oriented after the table of “TRL Definition and Decomposition by Factor” published by Frerking and Beauchamp in 2016 [43]. The difference in TRL between methods can be used to assess investments needed for establishing a particular method for the plastic sorting industry. The lower the TRL, the more investment is expected. The selection and ranking of the sorting technology were not based only on the TRL but also on the ability of the technology for high-throughput detection as well as polymer and marker identification (in the best case, of all polymer types at the same time).

A large number of potential materials was limited by further relevant parameters to obtain a reasonable number of materials at the end of the assessment. Concerning POM, a decision tree displayed in Figure 2 was followed based on the market availability of a tracer and therefore the “top candidates” or chemical substances are available for purchase and have a registration number according to REACH. The REACH authorization for the EU market is generally mandatory, at which safety-relevant data on their physicochemical properties needs to be provided when the production or import volume onto the EU market rises above 1 ton per year. If the production or import volume is >10 tons, >100 tons, or >1000 tons per year, an increasing number of additional toxicity tests are required for hazard and exposure assessment according to REACH [45]. It has to be highlighted, for nanomaterials, which are defined by the size range between 1–100 nm, more safety-relevant information is needed for the REACH registration [46]. These considerations are important for tracer substances that are still in the research and development (R&D) pipeline, as such substances would most probably exceed these REACH regulatory thresholds in the case of industry-scale implementation of plastic markers and would consequently require updates of information for the REACH registration. Subsequent selection criteria were mainly based on physical and chemical properties that are relevant for the processing of the additives into a polymer matrix, such as the melting and decomposition temperatures and the density. In addition, toxicity parameters were included for the selection in form of the hazard statements assigned to a substance according to registration in the Globally Harmonized System (GHS) or the REACH and CLP regulation. There would be other selection criteria or key performance indicators, such as costs, etc., but tracer costs in particular were not considered in this study because this would lead to the immediate exclusion of the relatively expensive tracers in innovative R&D projects.

## 3. Detection Techniques for Sensor-Based Sorting

Spectroscopic methods that are used or tested for industrial application of automatic sensor-based sorting systems in recycling plants can be based on color, X-ray fluorescence (XRF), near-infrared (NIR), or visual spectroscopy (VIS) sensors [12]. These sensor-based separation technologies are implemented to increase the efficiency in mechanical pre- or post-treatment of, e.g., lightweight packaging waste [27]. Spectroscopic methods are used either for sorting certain commodity plastics, i.e., sorting of PVC using XRF through the distinctive emission peak of chlorine in the XRF spectrum from the mixed plastic streams [27]. Similarly, in the case of RAMAN-spectroscopy, it is used for sorting multiple commodity plastics such as PP, PS, and ABS granules or flakes originating from household WEEE stream, in a high throughput setting [19,47]. Table 1 summarizes spectroscopic methods that either are already in use or are still in development for sensor-based sorting of plastic wastes, indicated by different TRL levels. In some cases, also the determination of the fraction of a polymer in a plastic can be of interest, which can only be achieved by quantitative analysis. Quantitative analysis via spectroscopy is defined as a method to assess the quantity of the target analyte in a sample often correlating with the intensity of analyte-specific peaks in the spectrum, whereas qualitative analysis via spectroscopy is defined as identifying the target analyte often using characteristic peaks in the spectrum as reference [48,49]. The measurement range depicted in Table 1 shows typical spectral ranges for detection, which may be adjusted depending on the instrument setup. Table 1 also summarizes examples of applications, specifically applicable to TBS. Other innovative detection methods, such as hyperspectral imaging as shown, e.g., on brominated plastic wastes from WEEE or post-consumer packaging materials, may be applicable for TBS but were not included in Table 1 because no application studies using tracers could be found.

### 3.1. Infrared Spectroscopy

Near infrared (NIR) and mid-infrared (MIR) spectroscopy are state-of-the-art detection methods used for coarse as well as fine sorting of polymers in waste recycling plants, but are only limited applicable for black polymers due to the strong absorption of infrared (IR) radiation by added black colorants [50]. Since the wavelength of IR spans over a large spectral region, from 0.78 µm to 1000 µm, and different technologies are used, it is generally divided into three regions: near-infrared (NIR = 0.78–3 µm), mid-infrared (MIR = 3–50 µm), and far-infrared (FIR = 50–1000 µm) [51]. Spectroscopic methods using different IR regions also require different instrumentations, and efforts for sample preparation, and display different characteristics in the resulting spectra [52]. NIR is, for example, used to separate packaging waste or even WEEE and differentiate PET, PE, PP, PS, PVC, etc., by an IR camera which is positioned above the convey belt and controls magnet valves for blowing out a polymer type under compressed air [12,16,53]. Both MIR and NIR systems can be modified for Fourier transformed (FT) measuring, which requires an additional component, an interferometer as well as compatible software. This will generate an interferogram which needs to be Fourier transformed to yield a spectrum [54]. In the field of polymer recycling, FT-IR is mainly used—in addition to NIR analyzers—for identifying different polymer types using desk instruments or portable handheld detectors to draw random samples for quality assurance, but not applicable for high-throughput automated sorting systems due to the relatively long duration of measurement and data processing. NIR represents a technology that is already used in the state-of-the-art sorting of plastics and could be used with appropriate NIR detectable markers (mainly up-conversion markers, which are explained in Section 3.2. to also separate polymers that cannot be distinguished by their intrinsic IR features alone.

### 3.2. Up Conversion Fluorescence Spectroscopy

Using up-conversion (UC) fluorescence spectroscopy, emission (fluorescent radiation) with a lower wavelength and higher energy level than the excitation radiation is created, therefore the most common excitation wavelengths being used are in the IR range [55,56]. Up-conversion fluorescence is a deviation from standard downshift fluorescence spectroscopy since the generated fluorescence is classified as anti-stokes. In the up-conversion process, two or more photons combine into one photon with higher energy than the original photons [57]. A multitude of different compounds are being developed for the use in up conversion fluorescence spectroscopy: For example, “doped” lanthanoid-based compounds (e.g., Y_2_Ti_2_O_7_:2%Yb^3+^, 1%Er^3+^), where the trivalent lanthanoids (Yb^3+^ and Er^3+^) are excited by the excitation photons and transfer energy to the lanthanoid in the host matrix (Y_2_), which in turn emits the detectable fluorescent signal [57,58]. A converter module containing periodically polarized non-linear crystals can be used to increase the photon energy through up-conversion which converts MIR radiation to NIR radiation to enable a higher detection rate but keep the higher resolution of MIR excitation radiation [52]. Up-conversion needs high photon densities (usually provided by lasers) and exhibits an excellent signal-to-noise ratio. However, background signals in the spectral range of the emitted photon can interfere with the detection. UC-spectroscopy was applied at pilot-scale investigation, where UC-tracers were printed onto PE-HD labels or incorporated into the matrix of blow extruded PE-HD bottles [29]. The technology could be used in the future for the detection of various polymers, but research is needed on the suitable up-conversion marker. Another limitation is that the detection signal is visual and therefore the detection needs to be performed in the dark because of the background visible light from the environment. Additionally, operating high-intensity open laser beams or pulses demands laser safety considerations.

### 3.3. X-ray Fluorescence (XRF)

XRF uses X-rays (spectral region 10 pm to 10 nm) to excite electrons within the inner shells of an atom and detects fluorescent X-rays, that are specific for each chemical element [59]. During stabilization and electron transition, fluorescent radiation is emitted (secondary X-ray), usually in a range of 200 pm to 60 pm and measured by a detector. Although theoretically applicable for any element, in practice, XRF is only suited for the detection of heavier elements, which is why authors recommend tracers with an atomic number larger than 29, when the detection is operated in the air [31]. For this reason, XRF is currently used for the detection and separation of PVC or plastics that contain brominated flame retardants because the chlorine or bromine peak, respectively, is very pronounced [10,15,33]. Industrial sorting systems using XRF are already available for recycling (e.g., Redwave XRF [10]) and can also be used to identify metals but are not as commonly used as IR-based polymer sorting systems. Additionally, X-rays are highly energetic and have to be blocked according to radiation safety recommendations [60].

### 3.4. UV-Vis and Fluorescence Spectroscopy

In UV-Vis spectroscopy, the absorbed energy of photons that is needed to lift ground state valence electrons in a material are measured, while the energy emittance of excited state electrons that fall back to ground state in dependence on excitation wavelength is measured in fluorescence measurements [61,62]. Since UV-Vis spectroscopy and fluorescence use high energy for excitation, high energy input for a high-throughput of marked polymers is needed, which can also lead to a bleaching of colored polymers [31]. Information found in literature suggests that UV-Vis spectroscopy is mostly used to identify certain polymer blends or sort out different colored materials [63]. Research groups have used fluorescence spectroscopy for TBS by introducing lanthanoids or different pigments into the polymer matrix at ppm levels [11,31,33,64,65]. However, in UV-Vis and fluorescence technologies, the spectral background in the recycling facility plays a major role and has to be considered when the instruments are calibrated.

### 3.5. Time-Gated Fluorescence Spectroscopy

Time-gated fluorescence spectroscopy (TGFS) is a supplementary method to other sensor-based detection methods, based on principles already used by time-resolved fluorescence spectroscopy (TRFS) [66]. TGFS assesses the difference between the decay time of the autofluorescence of the polymer materials and the decay time of the marker substance and through exact timing of the point of detection, the autofluorescence of the material can be almost completely excluded [66,67]. This more sophisticated version of fluorescence spectroscopy allows to reach better spectral separation when polymers or background exhibit autofluorescence and are therefore more accurate in identifying polymers. This concept can be applied to IR, RAMAN as well as UV-Vis radiation with the requirement that a fluorescent tracer or target material has a long fluorescent lifetime and hence a long decay time [64,66,68]. The development of TGFS has been pushed by the technological advancements of light sources, imaging and sensor hardware, and prototypes have demonstrated a relative high sorting quota of differently colored and marked POM but the technique is not yet ready for industrial usage since parameters like line speed, which correlates with the detection time, need to be improved [66,69]. 

### 3.6. Raman-Spectroscopy

In contrast to fluorescent spectroscopy, Raman spectroscopy relies on the inelastic light scattering properties of the target substance or material. The inelastic loss of energy is calculated to an IR spectrum. Simplified, scattered light can be divided into Rayleigh scattering where the scattered light has the same wavelength as the incoming light, and Raman scattering where the scattered light has either a higher or lower wavelength compared to the incoming light [70]. Raman spectra can provide sharp IR spectral peaks, due to the signal intensity being proportional to the excitation light intensity, which can be tuned by selecting different high-powered lasers but caution needs to be taken as high powered radiation may damage the target material, such as shredded polymer flakes (PP, PS, ABS) from WEEE [19,47]. The main advantages of Raman spectroscopy are that the technology is already being tested in an on-line scenario in some recycling facilities and that the technology is also usable for a marker-free detection. This makes it complementary to other sensor-based sorting techniques and TBS [19,47]. Additionally, recent studies have tested the applicability of Raman spectroscopy for identification of nano and micro-plastic particles which are an increasing cause of concern in waste management [71,72].

### 3.7. Laser-Induced Breakdown Spectroscopy

Laser-induced breakdown spectroscopy (LIBS) is another method that is being tested for the industrial sorting of polymers. It belongs to atomic emission spectroscopy techniques and uses a high-powered pulsing laser that creates a plasma plume on the surface of the target material [73,74]. The radiation that is emitted during the cooling of the plasma plume can be analyzed for different compounds in the source material [75]. Simple LIBS identifies the atomic composition and allows for quantification of elements, but research has shown that combing LIBS with statistical methods like principal component analysis (PCA), partial least square regression (PLS), line regression, and usage of inert gas atmospheres allows the identification of distinct polymer types as well as organic additives [75,76]. On its own, LIBS allows for the identification of some polymers (e.g., PVC, POM, and PTFE) as well as additives like different colors or stabilizers based on inorganic compounds in the polymer, but the correct identification rate may differ because of the coloring of the polymers [75,77]. Since the method yields a spectroscopic fingerprint of the elements in the material machine, learning algorithms could be used in the future to use additives as an indicator for certain polymers [78,79]. Recent studies have shown that LIBS has a very high identification rate for different polymers including ABS, PVC, POM, PTFE, PA, and PUR but the speed of detection needs to be increased to achieve possible implementation in an industrial setting [75,77,78,80,81].

**Table 1 polymers-14-03074-t001:** Compilation of relevant information for the different spectroscopy methods regarding tracer-based sorting, including the working principle, measurement range, technology readiness level, pros and cons as well as the qualitative or quantitative measurement of the method.

Method	Principle	Qualitative vs. Quantitative	Detection Wavelength (nm)	TRL	Pros	Cons	Example of Use for TBS	References
Near infrared (NIR) spectroscopy	Vibration of atomic bonds due to change in the dipole moment, mainly overtones and combination bands	Qualitative	900–1700	9	Fast, low cost (most sorting facilities would not need to buy new equipment), almost no preparation of samples needed	Dark polymers cannot be reliably identified, troubles with polymer mixtures and additives	NIR without the use of up-converting materials has not been researched for tracer-based sorting in part because of its limitations with dark colored polymers	[50,53,70,82,83]
Mid-infrared (MIR) spectroscopy	Vibration of atomic bonds due to change in the dipole moment, mainly deformation, stretching, etc.	Qualitative	2500–16,000	9	Compared to NIR peaks in resulting spectra are more intense, less problems with black polymers	Additional sample preparation needed, high detection time, tight contact to sample needed, not yet suitable for high throughput sorting	Not yet usable for TBS because of limitations through sample preparation, detection time, and contact to sample	[52,84,85,86]
Visual (UV-Vis) spectroscopy	Reflectance or absorption of visible radiation depending on color of samples	Qualitative	500–700	9	Fast identification of different colored polymers	Cannot sort for polymer type if they display the same color or different additives	No research of visual spectroscopy for TBS found	[62,63,87]
X-ray fluorescence spectroscopy (XRF)	Disturbance of electron equilibrium using high energy radiation and detection of fluorescence emitted during restoration of electron equilibrium	Qualitative and quantitative	0.062–0.248	9	Fast, cheap, very suitable for “heavy” tracers (mostly inorganic), can identify presence of brominated flame retardants	Can only differ between PVC and PVDC and other plastics but not between all plastic families without tracer substances	Nd_2_O_3_, Gd_2_O_3_, Er_2_O_3_, Yb_2_O_3_	[10,31,32,33]
Fluorescence spectroscopy	Energy absorbance of ground state electrons of elements/energy emittance of excited state electrons in dependence on excitation wavelength and intensity	Quantitative and qualitative	400–700	7	Fast, suitable for tracer-based sorting using organic as well as inorganic tracers	No characteristic spectra for different polymers, high energy radiation may influence material properties	Rare earths doped with rare earth or metallic oxides doped with rare earths (e.g., Al_2_Ba_2_Mg_2_O_7_:Eu^2+^; Y_2_O_2_S:Eu^3+^)	[33,61,62,65]
RAMAN spectroscopy	Vibration of atomic bonds due to change in polarizability	Qualitative	2800–20,000	7	Fast, supplementary to many spectroscopy techniques like NIR or LIBS	Weak intensity, much noise from colored plastics	No research on RAMAN for TBS found	[19,47,70,79]
Laser-induced breakdown spectroscopy (LIBS)	Element analysis via plasma radiation	Qualitative and quantitative	200–975	5	Almost no preparation of samples needed, allows identification of additives	May damage surface through high powered laser; online speed not enough	Not specifically mentioned for TBs but since detection of single elements is the principle of the method and used for identification of additives in polymers, detection of specified markers should be feasible	[75,77,78,88]
Time gated fluorescence spectroscopy (TGFS)	Decay time of fluorophores and autofluorescence of host material	Qualitative and quantitative	Dependent on spectroscopy method used	4	Improving Signal to Noise ratio, complementary technique to other spectrofluorometric methods	Expensive (additional hardware and software needed); may be limited in throughput speed	Supplementary to whatever main detection method is used; suitable for lanthanoids because they tend to display longer fluorescence decay time, than the polymer	[66,67,68,89]
Up-conversion (UC) fluorescence spectroscopy	Combination of two or more low energy photons to obtain emission of a single higher energy photon	Qualitative and quantitative	575–3600	4	Enables usage of lower tracer concentrations as well as lower energy radiation, suitable for detection of black polymers	Production of tracer molecules; expensive and complex	Y_2_Ti_2_O_7_:2%Yb^3+^,1%Er^3+^, as well as other lanthanoid complexes	[52,57,58,90]

## 4. Tracer-Based Sorting for Specific Polymer Types

### 4.1. Case Study—Polyoxymethylene (POM)

For the implementation of the TBS concept in waste management, additional sensors may have to be installed in the recycling plants if the existing sensors cannot be used, and markers will have to be mixed into the polymers. Both improvements, sensors, and markers will lead to increased costs. In addition to economic aspects, technical factors may limit the use of certain tracers, such as: (1) unclear availability or criticality of the tracer material—e.g., when using a tracer that consists of critical raw materials and therefore can be found on the Critical Raw Materials (CRM) list of the European Commission, (2) possible cross-contamination of markers with other plastics during the recycling process, or (3) low recyclability of the tracer itself. POM has a high market value as the primary material, but POM waste represents a problematic material flow, e.g., in recycling plants for WEEE, for which the TBS could be applicable to enable the separation of POM-pure material flows in an automated way. It is stressed that mechanical recycling or re-granulation of POM is generally very difficult due to the poor miscibility with other polymer types (cf. Appendix A). In addition, POM recycling is hampered also due to its small market share and waste volumes compared to, e.g., PP, PE, or PET, for which recycling technologies have been implemented and recycling markets have already been established. According to market analysis, POM comprises together with a few other special-purpose polymers, i.e., polybutylene terephthalate (PBT), acrylonitrile styrene acrylate (ASA), ethylene propylene diene monomer rubber (EPDM), 3.63 million tons or only 7.4% of the total polymer converter demand in the EU (EU27 +3) [2]. However, semi-/finished plastic products made of POM polymer have certain advantages over using metals for the same function, such as design flexibility, high strength-to-weight ratio, lower costs, corrosion resistance, etc., but also over other similar thermoplastic polymers, such as good strength and impact resistance, hard surface with good appearance, excellent dimensional and chemical stability [91]. For this reason, POM materials are used as components in products for many different sectors and products. Examples of applications can be taken from Appendix A.

POM plastic is mostly used in the electronic, construction, and automotive sectors and consequently, POM wastes follow several different disposal routes (Figure 3). For example, POM plastics from WEEE or end-of-life vehicles are not separately collected and shredded with the main components during waste pre-treatment. These POM material streams have the highest potential to be recycled using automated sorting systems detecting one of the tracers that are proposed in Section 4.2. POM plastics from household items and textiles are disposed of either directly in the residual waste and used mostly for energy recovery or collected with other recyclables, where they will mostly end up again in waste incineration due to their poor detectability and low quantity compared to other recyclable polymers. POM plastic from the construction and furniture industry, unless collected as a completely separate waste stream, has a similar disposal route to household utensils and textiles, and will mostly end up in waste incineration. An overview of the overall POM value chain in the case of the EU is shown in Figure 3. In summary, closed-loop recycling is currently not feasible and therefore, we propose to introduce a tracer and may implement the TBS concept.

### 4.2. Selected Tracers (Three Promising Candidates)

The selection procedure we developed narrowed the list of tracers suitable for the TBS concept from 80 compounds (Appendix A) to 3 promising candidates. These are cerium (IV) oxide (CeO_2_), yttrium (III) oxide (Y_2_O_3_), and perylene-3,4,9,10-tetracarboxylic dianhydride (PTCDA). CeO_2_ and Y_2_O_3_ are oxides of rare earth materials that do not melt below 400 °C and are therefore stable during extrusion. CeO_2_ has a higher density than Y_2_O_3_ with 7.2 g/cm^3^ compared to 4.85 g/cm^3^, which could lead to different influences on material properties when added in higher concentrations. Both lanthanide-oxides have no assigned hazard statements and are not classified under GHS according to the information and tests provided for REACH registration under which CeO_2_ is registered with production or import volumes of >1000 tons per year and Y_2_O_3_ with >100 tons per year. However, producers and importers have assigned nine different hazard statements to CeO_2_ in total and four to Y_2_O_3_ in accordance with CLP. According to the selection criteria postulated by different research groups, both lanthanoids fulfill the criteria of availability, safety, and suitability as marker substances for different spectroscopic methods like XRF and UC-IR as well as LIBS [31,33].

PTCDA is an organic molecule and a perylene pigment that is already in use for polymer coloring with a predicted melting point above 500 °C, which would also make it thermostable during extrusion. It has a density of 1.684 g/cm^3^, which is considerably lower compared to selected lanthanoids. PTCDA is produced or imported in a volume of >1 ton per year, has no hazard statements attached, and is not classified under GHS according to its REACH registration. Following CLP, it has been assigned three different hazard statements by producers and importers.

Revisiting the information from the overview of spectroscopic methods (Table 1), CeO_2_ and Y_2_O_3_ should be detectable in polymers or plastic wastes using all methods that can be used for TBS, while PTCDA was primarily chosen for IR spectroscopy. The next steps would be to introduce these selected tracers into POM or other “problematic” plastic wastes in different concentrations, conduct material testing, and experimentally prove their applicability and detectability at the pilot scale, but these empirical investigations went beyond our study.

## 5. Discussion

TBS has the potential to significantly increase the recycling efficiency for certain polymer types, but to enable a smooth transition from an established recycling system to a newer system, a lot of different aspects have to be considered and addressed. TBS will require investments and therefore, its implementation needs to be evaluated for feasibility, profitability, and sustainability of the technical systems in the economic, social and political systems, and the affected stakeholders.

The transition from a linear value chain to a circular economy requires major efforts in terms of sustainability, which has already been discussed by research groups Brunner et al. [11] and Bezati et al. [31]. In the case of lanthanoids, the natural availability has to be considered for avoiding a shortage of materials (supply risks) that may occur if they are to be established as tracers on an industrial scale. In addition, the mining and production process needs also to be considered during the sustainability assessment. The goal for TBS, regardless of the tracer compound used, should be strictly adhering to the principles of green chemistry during polymer production [92]. This would include a high recovery and reuse of tracer compounds to limit the need for new tracer material and thus depletion of natural reserves of such critical raw materials. Bezati et al. calculated a need of 400 tons of each tracer substance per year for one or more polymers that are produced in a total volume of 1 million tons per year if three primers are used in binary combination. Prior research found that the concentration of marker incorporated into the polymer matrix should revolve around 100 ppm or 0.1 g per kg polymer [29,31,33]. This consideration is based on two principles: first, using the minimum amount of marker that still enables a robust detection and second, maintaining the material properties of the plastic that is labeled by the marker. Research using PP and ABS as host polymers has shown that a lanthanoid-doped lanthanoid-oxide marker complex, as well as a lanthanoid-doped metal-oxide marker complex concentration of up to 250 ppm, did not influence material properties such as traction, impact strength, and flexure [33].

The usage of a variety of tracer substances and different successively applied methods of detection and separation may benefit from the establishment of a tracer coding for different polymer products/compositions which would need to be implemented into international regulations. As a first step, the implementation of TBS may be reasonable when it is introduced for certain valuable polymers or for polymers that reduce the quality of recycled materials when they enter the recycling process, such as POM. The industrial implementation of the TBS concept would also require a sophisticated data management system. For example, spectroscopic methods need reference spectra, tracer development, and selection needs material- and safety-specific information, etc., which must also be passed on to third parties, whereby no intellectual property rights should be violated. Other research groups, such as Ahmad and colleagues [30], have also created a database for their tracer substances and their database could be merged with ours. The establishment and maintenance of a marker database on a multinational level would allow for easy access to required information for researchers and producers especially if a “binary code system” with multiple tracers is to be established. To include sorting-service providers in the usage of the database, reference spectra of marked polymers should be included in the database with the corresponding binary code identifying the correct polymer product. First data collection could be made during a subsidized research project.

Gasde et al. highlighted economic drivers and barriers for TBS, which might be viewed by many as a radical method, able to drastically change established production and sorting systems. The study showed that a big part of relevant stakeholders in polymer recycling, including technology developers and providers, legislative bodies, recycling businesses, packaging producers and others, had the biggest concerns about “regulatory and legal barriers”, “distribution of efforts/benefits”, “profitability and competition”, “quality and safety concerns” and “process and technical concerns” [18]. Keeping those in mind, the establishment and maintenance of a database would at least provide advantages in the category of “distribution of efforts/benefits” by providing basic data. If the data is implemented into official and managed databases that are publicly accessible, e.g., REACH registration dossiers, it would guarantee a certain quality of data and address “quality and safety concerns”.

To address “process and technical concerns” as well as “quality and safety”, the TRL of sensor-based sorting methods as part of the TBS can be used. A higher TRL requires more research, which in turn means that a lot more knowledge has already been generated and safety and security concerns will most likely have already been eliminated in order to pass to the next level of the TRL. Even though the estimated TRL for XRF and NIR are the same, systems that use XRF as one of the sensor-based sorting methods are not as widespread in use as NIR systems. This is most likely caused by the missing capability of the method to accurately identify polymer groups aside from PVC without marker materials and therefore not generating enough scientific and industrial interest in polymer sorting to achieve high industrial use. Nonetheless, recent research on lanthanides as suitable markers for TBS, as well as the advantage of XRF being a non-contact and non-destructive detection method, has increased interest in the potential industrial use of XRF. Both methods can identify lanthanide markers either through up-conversion in the case of the NIR or standard down shift spectroscopy in the case of XRF. This presents certain advantages, since the most suitable lanthanides are well researched, with data on toxicology, and physicochemical parameters, and are registered under REACH. Coupled with the high TRL of both detection methods, they present a desirable economic opportunity to the other reviewed methods like LIBS with lower TRL or RAMAN which has currently not been applied for the TBS concept.

Considering a whole polymer recycling system of different sorting methods, it seems most plausible to establish a line-up of conventional detection methods and TBS to achieve the most efficient result targeting recycling rates, carbon footprint, and other key parameters. The combination of different detection methods (e.g., RAMAN and NIR spectroscopy) either consecutive or through sensor fusion is already being tested by different working groups [79,93,94] and might reach a feasible TRL in the near feature. This would also coincide with one of the key challenges found by Gasde et al., which is the compatibility of the new TBS methods with established sorting methods. Gasde and colleagues assessed two different scenarios for implementation of TBS into already existing sorting pathways: In the “TBS light” scenario, TBS sorting machines are added at the end of conventional sorting processes for conventionally “pre-sorted” plastics to increase the sorting quality of the waste streams by removing problematic material like multilayer material from existing polymer streams. In the “TBS complete” scenario, conventional sorting processes are replaced by TBS methods to establish individual pure polymer waste streams, including different sub-streams of identical polymers with different parameters (e.g., food vs. non-food contact material) [18,29].

It is expected that the implementation of “TBS light” is more likely due to the lacking flexibility of the established recycling system including multiple stakeholders, as “TBS-complete” would need a radical change in the existing system that comes with large investments and changes in the regulatory landscape and affects industries along the whole value chain. At the same time, “TBS light” may not yield the desired increase in recycling rates and reduction in the carbon footprint required to achieve the goals set in the circular economy plan by the EU [9,18,27].

To provide incentives for companies to invest in a TBS system, it has to be profitable. Costs are generated through various sources, such as development and acquisition costs for the sorting systems, material costs of marker substances, registration costs for REACH registration (if an unregistered substance is used), and so on. The maximum economic-compatible cost for a tracer is hard to estimate since parameters such as the value of the recycled polymer, development and production costs of the tracer, registration fees, and taxes need to be considered. For example, in 2021 the EU introduced a “plastic tax” of EUR 0.80 per kg of non-recycled plastic that needs to be paid into the annual budget of the EU. This allows the maximum costs of tracer to rise and still be economically advantageous, as the cost in taxes will be reduced by increasing the recycling rate through the implementation of the tracer.

A registration under REACH becomes necessary if the imported or produced annual tonnage reaches one ton. This is the case when 10,000 tons of polymers are marked using 100 pm of marker per kg of polymer, since the total volume of the marker needed would equal 1 ton. A REACH registration that must be filed introduces additional risks and market barriers. According to a survey about REACH registration, the chemical and toxicological analyses, including documentation to fulfill the REACH requirements for a 100–1000 tons per year, costs on average around EUR 800,000 in the EU or Switzerland [45,95]. Looking at the market data provided by PlasticsEurope on the state of plastics demand in the EU 27 + 3, with PP having the highest demand with 9.7 million tons, and PMMA the lowest with 0.26 million tons, suggests that even when markers are introduced to polymers with lower production or import volumes which are summarized within “other plastics”, a total demand of 3.65 million tons will most likely require REACH registration of the marker substance. Therefore, substances that are already filed in REACH are economically advantageous over materials that still must be registered. An existing REACH registration also mitigates the risk that the substance shows unexpected safety concerns during the testing and becomes ineligible for the application as a marker. Additional investment costs to upgrade the machinery for processing the tracer materials and the costs in particular to implement detection technologies need to be considered—in general, the costs grow with each TRL [96,97]. 

## 6. Conclusions

There are already several spectroscopic methods that are applicable on an industrial scale for automated sensor-based sorting of various plastic streams. The most widely used method is NIR spectroscopy due to the high throughput capacity, high sorting efficiency, and relatively low investment costs, but it is reaching its limits with dark-colored polymers and polymer composites. Besides NIR-based detection techniques, UV-Vis and XRF are also useful for TBS applications with some limitations, particularly regarding throughput capacity and relatively high costs. In principle, UV-Vis can distinguish plastics by color, but not by polymer type, and XRF can identify only PVC or other polymers that contain halogens or other additives by detecting chlorine or other elements with an atomic number higher than 11 (sodium). All three techniques can be used to distinguish polymers that are labeled with tracers.

This review shows that selecting appropriate tracer materials is a complex procedure that needs to consider the material compatibility of the tracer and the polymer type, availability of the tracer, the detection technology, and the limit of detection, as well as safety, regulatory, and economic aspects. A general consideration of these aspects led to the finding that PTCDA, CeO_2_, or Y_2_O_3_ are promising tracers that can be used to label special polymer types such as POM with a relatively high market value. From a theoretical consideration, the TBS concept appears to be generally feasible for certain problematic polymers in mixed plastic waste streams, such as plastics from WEEE recycling. However, such cases need to be studied in more detail in the framework of feasibility and implementation studies that must prove if TBS is improving the homogeneity of a recycling stream and increasing material quality of recyclates sufficiently to outweigh the additional investment and implementation costs for a TBS system. Although TBS is mostly viewed as not economical at present, the change in regulatory framework could make TBS a promising technology for certain plastic products, such as POM or other valuable thermoplastics, to achieve higher recycling rates in the future.

## Figures and Tables

**Figure 1 polymers-14-03074-f001:**
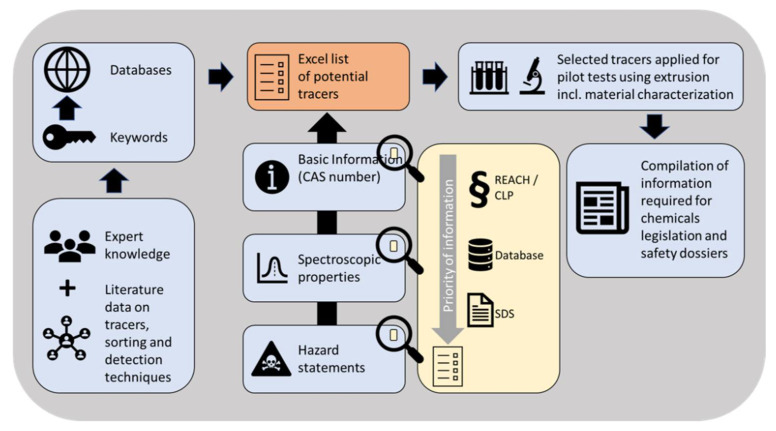
Scheme of the workflow to select a suitable substance that qualifies for tracer-based sorting of plastic wastes. The yellow rectangle displays the priority of information, as in best case that the tracer is commercially available, indicated by the CAS number, and safe (not persistent, not bio accumulative, and non-toxic) according to the REACH and CLP regulation summarized in safety data sheets (SDS).

**Figure 2 polymers-14-03074-f002:**
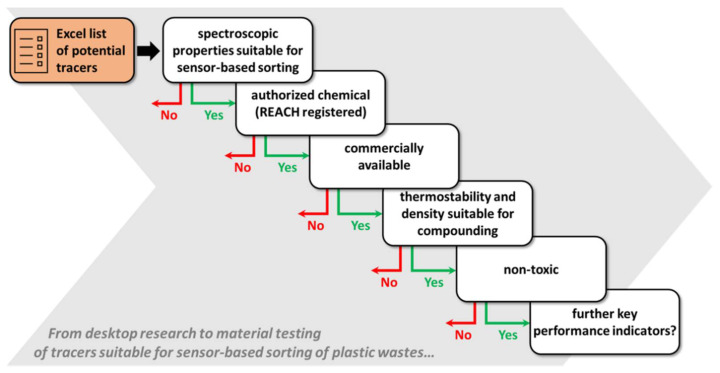
Proposed decision tree for selecting tracers from the compiled excel sheet (Appendix A). Parameters can be switched or modified according to requirements. Costs of the tracer substances were not considered in this study.

**Figure 3 polymers-14-03074-f003:**
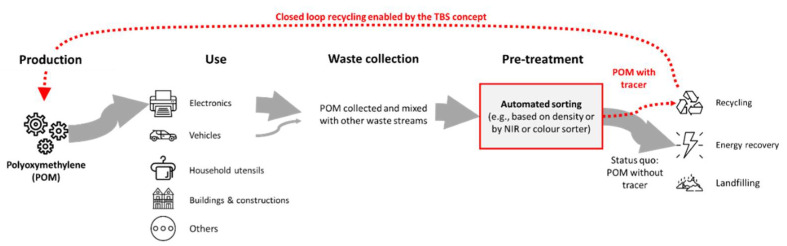
Overview of production, use, and disposal routes for POM plastics in the EU. Recycling efficiency for POM waste may be significantly improved by applying the tracer-based sorting concept.

## Data Availability

The data presented in this study are available on request from the corresponding author.

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
