# Peer review of "Evaluation of Marker Materials and Spectroscopic Methods for Tracer-Based Sorting of Plastic Wastes"

_polymers, 2022, doi:10.3390/polym14153074_

Round 1

Reviewer 1 Report

The current manuscript entitled “A methodical approach to the selection of marker materials for tracer-based sorting of plastic wastes” by “Olscher et al” reviewed and assessed plastic sorting techniques and spectroscopic detection methods, applicable to tracer-based sorting. Furthermore, a comprehensive list of potential tracer substances suitable for thermo plastics was derived from the literature. Finally, we developed a set of criteria to select the most promising tracer candidates (3 out of 80) based on their material properties, toxicity profiles, and detectability that could be applied to enable the circularity of, for example, POM plastics. Manuscript seems good, interesting and written well. The article can be accepted after addressing the following comments.

1.       Please remove the abbreviated forms in the keywords section

2.       Give the full forms of PET, PP, PE, or PS, in the abstract section

3.       Lines 57 and 58 provide the full forms.

4.       In the abstract section, line 15, authors stated that; while many engineering plastics, such as polyoxymethylene (POM), ………… Provide some more (name of plastics) engineering plastics.

5.       Check line number 189, Reference is missing

6.       On page number 6; figure caption is missing for the figure.

7.       From the lines 208-210; connectivity in between the lines is missing. Please cross check.

8.       Superscripts are not properly given in the manuscript; for eg see lines: 293

9.       Conclusion should be written as conclusions

10.   The authors should cite the following articles on recycled plastic waste

https://doi.org/10.3390/polym14102038

https://doi.org/10.1016/j.jclepro.2022.132532

https://doi.org/10.3390/polym14112275

Reviewer 2 Report

1-    The abstract must be included with some important data from this research.

2-    In the abstract for PET, PP, PE, and PS, first write the full form and then use the abbreviation.

3-    Line 45, approx. 41 % (write the complete word)

4-    Line 57: PE-HD, PE-LD, PE-LLD, PE-MD write the complete form first.

5-    The last paragraph of the introduction must be specified for the purpose of the study.

6-    Line 189: what is this message? “Error! Reference source not found”

7-    Section 2.2., why the figure is replicated in this section without any caption?

8-    Separate the discussion and conclusion parts. Don’t use any references in the conclusion section and write the conclusion in your own words.

9-    Write a future prospect at the end of the manuscript.

10-   Use the following references in the manuscript:

Muhamad, I. I., Sabbagh, F. A. R. Z. A. N. E. H., & Karim, N. A. (2017). Polyhydroxyalkanoates: A valuable secondary metabolite produced in microorganisms and plants. Plant Secondary Metabolites, Volume Three: Their Roles in Stress Eco-Physiology, 185.

Sabbagh, F., Muhamad, I. I., Pa’e, N., & Hashim, Z. (2018). Cellulose-Based Superabsorbent Hydrogels.

Reviewer 3 Report

The authors presented an interesting manuscript, but there issues that need to be addressed before approval.

English writing is sound.

References are not within format.
There are referencing errors throughout the document (clearly presented in lines 194 and 216).

The presentation as an original article is odd as most of the development is based on literature review. For a review article, the present format would be ok, but the number of References is somewhat short for a document of this type. For an original article, the authors must clearly present the Methodology and Results of their work.

The novelty is obscure both on the Asbtract and Introduction.
The Objectives are not clear after the Introduction.

Round 2

Reviewer 3 Report

The authors improved their manuscript. It has the capacity to be an interesting reference in the area for the nex years.